# Museum Moving to Inpatients: *Le Louvre à l’Hôpital*

**DOI:** 10.3390/ijerph16020206

**Published:** 2019-01-13

**Authors:** Jean-Jacques Monsuez, Véronique François, Robert Ratiney, Isabelle Trinchet, Pierre Polomeni, Georges Sebbane, Séverine Muller, Marylène Litout, Cécile Castagno, Didier Frandji

**Affiliations:** 1Cardiology, Policlinique Médicale, Hôpital R Muret, Hôpitaux Universitaires de Paris Seine Saint Denis, F-93270 Sevran, France; 2Department of Geriatric Medicine, Hôpital R Muret, Hôpitaux Universitaires de Paris Seine Saint Denis, F93270 Sevran, France; veronique.francois3@aphp.fr (V.F.); georges.sebbane@aphp.fr (G.S.); 3Department of Pharmacy, Hôpital R Muret, Hôpitaux Universitaires de Paris Seine Saint Denis, F-93270 Sevran, France; robert.ratiney@aphp.fr; 4Department of Addictions, Hôpital R Muret, Hôpitaux Universitaires de Paris Seine Saint Denis, F-93270 Sevran, France; isabelle.trinchet@aphp.fr (I.T.); pierre.polomeni@aphp.fr (P.P.); 5Musée du Louvre, F-75001 Paris, France; severine.muller@louvre.fr; 6Board of Directors, Hôpitaux Universitaires de Paris Seine Saint Denis, F-93000 Bobigny, France; marylene.litout@aphp.fr (M.L.); cecile.castagno@aphp.fr (C.C.); didier.frandji@aphp.fr (D.F.)

**Keywords:** art therapy, museum, hospital-related stress

## Abstract

Anxiety and depressive symptoms are common in hospitalized patients. Arts and cultural programs were reported to enhance their quality of life. The *Le Louvre à l’hôpital* study presents a new approach in which the museum moves to the hospital by displaying and discussing artworks with patients interactively. Over one year, four large statues were disposed in the hospital gardens, 30 reprints of large painting were exhibited in the hospital hall, dining rooms, and circulations areas. A total of 83 small-group guided art discussions (90 min) were organized, which 451 patients attended. The 200 small-size reproductions of paintings placed in the patients’ rooms were chosen based on their individual preferences. Decreased anxiety after the art sessions was reported by 160 of 201 patients (79.6%). Out of 451 patients, 406 (90%) said the art program had met their expectations, and 372 (82.4%) wished to continue the experience with caregivers (162 paramedics trained for art activity during 66 workshops). In conclusion, moving the museum to the hospital constitutes a valuable way to provide art activities for inpatients in large numbers, which may reduce hospital-related anxiety in many instances.

## 1. Introduction

Clinically important posttraumatic stress disorder symptoms have been reported to occur in a substantial proportion of critically ill patients discharged from intensive care units (ICU), of survivors of severe sepsis, and of patients hospitalized for cardiovascular events or for other serious medical diseases [1,2]. Depressive symptoms are also common during and after their in-hospital stay [3,4], as well as they are in elderly admitted to acute care units, rehabilitation units, and nursing homes [4,5]. While such depressive symptoms are often unrecognized, undiagnosed, and therefore untreated [6], they are associated with increased risk of hospital readmission, higher outpatient service utilization during follow up, higher nursing home admission rate in elderly, and overall increased mortality rates [7,8,9]. On the opposite, screening for and treating depression during hospitalization can improve outcomes [8]. Several non-pharmacological interventions have previously been used to alleviate hospitalization-induced depression and anxiety. Early intra-ICU psychological intervention promotes recovery from posttraumatic stress disorders, anxiety, and depression symptoms in critically ill patients [10]. Arts and cultural programs, specifically participatory-based art interventions, may provide unique opportunities to enhance the quality of life of inpatients. Listening to music has been shown to have a beneficial effect on anxiety in patients operated on during the postoperative period [11], in those hospitalized for coronary heart disease [12] and for cancer [13]. Art therapy programs for older patients are also typically comprised of music, art, drama, movement, dancing, or mixed modes. However, very few studies have examined the potential role that cultural programs such as visiting art museums may play in improving quality of life of patients [14]. Further, previous studies reported health-based interventions in museums or art-galleries consisting of small group tours of ambulatory patients only, but this was intended mainly to patients hospitalized in geriatric rehabilitation units, nursing homes or Alzheimer clinics [14,15,16].

The present *Le Louvre à l’hôpital* study reports a new approach in which the museum moves to the hospital and to the patient himself by displaying and discussing on an interactive basis artworks in patients’ rooms, dining rooms, circulation areas, and hospital’s garden. Art-making activities included enquiry-based discussions with patients in small groups led by one of the museum’s education staff. The program also provided training for hospital paramedics and caregivers to subsequently replace the museum’s staff in managing groups of patients further.

## 2. Materials and Methods

### 2.1. Design and Setting

This art engagement activity was established through a partnership between the *Musée du Louvre*, and the *Assistance Publique-Hôpitaux de Paris (APHP)*. This activity was planned by the museum’s and APHP’s boards of directors to improve quality of stay of patients hospitalized in Paris and city’s suburban areas. Pro forma reprints and copies at a 1:1 scale of artworks from the Louvre museum, Paris were offered for exhibition in selected medical departments of René Muret hospital, a hospital belonging the Hôpitaux Universitaires of Paris-Seine Saint Denis group which is located in a large suburban area Northeast of Paris. Artworks consisted of 4 large monumental statues displayed over the hospital gardens, 30 copies of large masterpieces of painting displayed over the hospital hall, dining rooms, and in the circulation areas, 200 reproductions of small size paintings displayed in the patients’ rooms (Figure 1). Patients were asked for choosing paintings to be placed in their personal room on the basis of their own preference using a catalogue with selected reprints from the Louvre’s collections.

Artworks were specifically displayed in departments in which patients were assumed to be more prone to depression and to stay for time enough to benefit from the museum’s activities. This included the departments of rehabilitation (geriatric medicine: 3; day care unit for geriatric medicine: 1; nutrition-obesity: 1), the department of medicine of addictions, the department of long-term care geriatrics, and the department of palliative care.

Guided-small group tours for patients were organized to present and discuss masterpieces artworks, monumental statues in the garden, large paintings in the hall and circulation areas. They lasted for about 90-min and consisted in an interactive discussion on the artworks displayed in the light of patients’ personal cultural and emotional experience.

Small group guided discussions on art were also organized during tea or coffee-time within the included departments. They were conducted by a trained-facilitator of the museum‘s education staff using a screen to guide the discussion. These guided discussions lasted for 90 min and commented 6 to 8 artworks depending on the patients included, and their ability to maintain attention. Patients were encouraged to look at the artwork and describe what they saw, using selected questions to initiate interactive comments such as “*What word first comes to mind when looking at this artwork*?”. Though interactive the session should be prepared depending on the participants too. The museum staff member and the attending health care givers defined a theme to which talks should focus, movement, smiling, representation of children, women, meals, portraits, seasons, mythology. Patients with higher educational levels were proposed to talk about a specific period of arts, such as the Renaissance, or a single artist career and creation.

A few structured visits to the Louvre Museum were scheduled. They focused on the artworks whose reproductions were displayed over the hospital to show how the original artworks look and how they appear in the museum’s exhibition.

According to the partnership between the Louvre and the hospital, high-quality copies and replicas of art were made by the museum’s graphical department and thereby copyright-free. Making copies and replicas were at the museum’s expense, as were museum’s staff members conducting guided art discussion. Health care givers engaged in the program participated to the sessions during their working hours.

The program was approved including for ethical issues by the Comité Executif Local (CEL) of the Hôpitaux Universitaires de Paris-Seine Saint Denis.

### 2.2. Hospital-Related Anxiety

The subgroup of patients from the department of geriatric medicine was tested for anxiety before and after the small group guided art discussions organized as a coffee-break lasting for about 1 h. Since changes in anxiety associated with short punctual interventions are rather difficult to rate using conventional scales such as the Hospital Anxiety and Depression Scale (HADS) [17,18,19,20], patients were asked qualitatively for their perception of decreased anxiety after completion of the session. The following items were recorded: Did this session provide any relaxation or feeling of well-being? Did you feel some improvement of hospital-related anxiety during the session?

### 2.3. Satisfaction Survey

A satisfaction survey was designed to assess participants’ overall satisfaction with the program, including guided-tours, museum visits, and small group discussion focusing on a painting or artwork. Patients were asked for satisfaction and whether they would enjoy participating again to a session to follow.

## 3. Results

### 3.1. Museum Activities

From May 2016 to May 2017, a Museum staff member was present at the hospital 40 full days during the 1-year program. A total of 83 small-group guided art discussions (coffee break or workshop lasting for 90 min) on 3 to 8 exhibited paintings were organized, to which 451 patients and 160 healthcare givers attended. Workshops to which patients and caregiver attended together allowed patients to interact more closely. Also, sessions reached more interactive results when focusing on concepts easy to understand for a broad assistance (Figure 2). This included studies on smiles, meals, family and couple life, children, or on very popular artists such as Leonardo da Vinci, or on worldwide known historical figures such as Napoleon, Cleopatra etc. Artworks were presented on A3 frames with several samples available (ratio 1 print for 2 or 3 patients). When a specific aspect was discussed, it was displayed using an enlarged scale on specific sheets, so the patient was able to touch the subject of the discussion, a smile, a meal, a child. Then, patients frequently commented on their emotional feeling while referring to their personal life history. They also engaged interactive views with other patients, the museum staff member or a paramedic asked to agree or disagree with the first patient’s perception.

All 200 small-size reproductions of paintings placed in the patients’ rooms were chosen by the patients on the basis of their individual preference. The artwork’s appreciation, visual appeal and choice were commented by the patient during an interactive talk with his regular healthcare givers and paramedics supervised by a member of the museum staff. All patients agreed for choosing and displaying an artwork copy in their individual room. There was no refusal. Also, all patients gave reasons for their choices. This lasted for a few words in some cases, but in most instances, patients reported more extensively their perceptions of the artwork, the reasons for their choices, and the personal life-recollections they advocated about.

In addition, 72 patients moved to the Louvre for a structured visit (total 14 visits) focusing on the artworks whose reproductions were displayed over the hospital, providing thereby information how the original artworks look and how they appear in the museum’s exhibition. These structured visits were relatively time-consuming, taking about 4–5 h each, due to the distance between the museum and the hospital. They required paramedics in substantial numbers, ranging from 1 for 3 to 4 patients, and, finally, provided less satisfaction to patients. Indeed, while most of them (62/72, 86.1%) enjoyed the visit, one half (37/72, 51.3%) said they were tired when back to the hospital, especially older patients. Finally, visits to the museum concerned 72 patients only while 451 patients attended museum activities inside the hospital, i.e., a 6-fold lower rate.

In parallel to the museum activities devoted to patients, 66 workshops and staff training sessions were held to establish further collaborations and to familiarize respective staffs and healthcare givers with the museum displayed activities. They were trained for continuing the art engagement activity with the patients in the years to follow. The museum curators and staff trained 162 paramedics using specifically devoted slide sets and short technical notices detailing methods to handle patients with further art activities. This is particularly relevant since paramedics reported that the Museum staff plays an important role in keeping the activities interesting and helping them to feel comfortable and accepted. During these sessions, paramedics were trained to continue the museum staff expertise in driving patients for art interactive discussions.

### 3.2. Anxiety

Among patients hospitalized in the departments of geriatric medicine and asked for anxiety by paramedics before and after a small-group guided art-discussion, 160/201 (79.6%) reported decreased anxiety after the session. Among patients hospitalized in geriatric medicine, there was no difference with regards to the improved anxiety depending on the type of stay. Patients admitted in the rehabilitation day care unit, in the rehabilitation unit and in the long stay care unit reported similar responses.

### 3.3. Satisfaction Survey

Among 451 patients asked for satisfaction with the art workshops and guided art discussions, 406 (90%) said the experience met their expectations and 372 (82.4%) said they would like to attend the program again. On average, as reported by the museum staff and paramedics attending the sessions, patients rated the small group sessions program highly, at 8 to 10 out of 10. With regards to patients’ satisfaction, there were no differences between those hospitalized in geriatric units and those hospitalized in the department of addiction and rehabilitation. Also, according to the museum staff and caregivers participating to the program, age, association of mild or moderate cognitive impairment, as well as the type of disease were not associated with differences in satisfaction of patients. Noteworthy, responses provided by younger patients from the department of addictions were as enthusiastic as those of older patients hospitalized in geriatric medicine.

## 4. Discussion

Art, music, cultural programs, and participatory-based art interventions have been shown to provide opportunities to enhance quality of life and to reduce hospital-related anxiety and depression in a large panel of inpatients [11,12,13]. Among these approaches, visiting art museums has previously been proposed, mainly for patients hospitalized in palliative care units or for elderly hospitalized in nursing home or day care units [14,15,16]. Unfortunately, such opportunities are not suitable for severely disabled patients or those with mobility impairments. This shortcoming has been overcome in the present study in which the museum moves to the hospital and to the patient. In this new approach, artworks were displayed over the hospital hall, gardens, dining rooms, and in the circulations areas, and in the patients’ rooms. Patients were encouraged participating to guided art discussions, to workshops, and to visits at the Louvre while not confined to bed. The program also included training sessions for healthcare workers to allow further continuation of the hospital’s art engagement after the study. The program was judged very satisfying by a large majority of patients. In a subgroup of patients hospitalized in the long term care department, art guided discussions reduced hospital-associated anxiety in a substantial proportion of patients, including those with mild Alzheimer disease.

Another new approach of this study consisted in promoting self-choice of the artworks displayed in the individual rooms. Reproductions of paintings displayed over the patients’ rooms were chosen by the patient himself or herself on the basis of personal appreciation, visual appeal, and preference. Although understanding of the experience of beauty and aesthetics has been reported in many aspects elusive [21], we were surprised hearing patients speaking about their perceptions of the artwork, the reasons for their choices and the personal life-recollections they advocated about.

Reasons for the potential beneficial effects of art interventions remain unclear. However, there seems to be evidence that actively participating could have better healing effects [14]. Also, stimuli with significant configurations are also ones that are more potent in recruiting attentional mechanisms and therefore attracting attention than stimuli that lack them and are therefore more neutral [21]. Aesthetic experience in the visual arts has neural correlates in the human brain. Studies that used patient-selected music resulted in greater anxiety—reducing effects [12]. Indeed, humans display a consistent preference across various images and artworks. Preference for visual patterns might be related to their sensitivity for such patients [22], since vision is an active process depending as much upon the operations of the brain as upon the external aesthetic appeal of artwork displayed. Assigning preference ratings has been shown to be independent of the ability to recognize the artworks on which the preference rate has previously been given in a memory task [16]. *“The art of the receptive field may thus be defined as that art whose characteristic components resemble the characteristics of the receptive fields of cells in the visual brain and which can therefore be used to activate such cells”* [21,23]. Accordingly, a museum therapy program in the hospital rather look as a specific therapy targeted to a specific patient than as a nonspecific approach devoted to improve quality of life during hospital stay only.

Finally, the training program devoted to healthcare givers and paramedics provided an unrivalled opportunity to optimize further maintenance of the museum art engagement program inside the hospital for the years to follow. Whether this approach could fulfill such satisfying achievements in the future remains, obviously, uncertain. However, the first steps have been completed to make this possible.

### Limitations of the Study

One major limitation to the study pertains to the lack of control group. Peer counseling might have been as effective in improving patients’ anxiety as art intervention, especially while helping them speaking about their life and personal recollections. However, this approach may rather recall perceptions of unpleasant changes in life among elderly or severely disabled patients, as well as reactive drug-addiction related anxiety among drug-addicts. Conversely, patients wanted to speak about their perceptions of art, the reasons of their choices and the personal life-recollections they advocated about, and this process was not associated with any reactivation of the disease which caused the hospital stay. A second limitation relates to the lack of follow-up assessment after the art therapy ended. However, trained paramedics performed subsequent art therapy sessions with patients using art sets from the Louvre and discussions about painting replicas which remained in place after the program ended. Finally, art therapy may not be useful to patients who don’t enjoy art. However, there were no refusal among patients asked for participation to the program.

## 5. Conclusions

Moving the museum to the hospital is a new modality of art-based therapy. It constitutes a valuable way to provide art activities for inpatients in large numbers. Such an approach is judged very satisfying by patients and may, in many instances reduce hospital-related anxiety.

## Figures and Tables

**Figure 1 ijerph-16-00206-f001:**
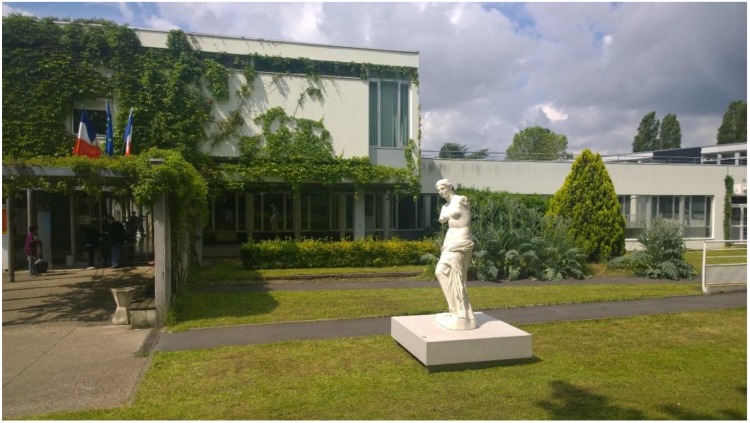
The copy of the Venus de Milo in the hospital garden.

**Figure 2 ijerph-16-00206-f002:**
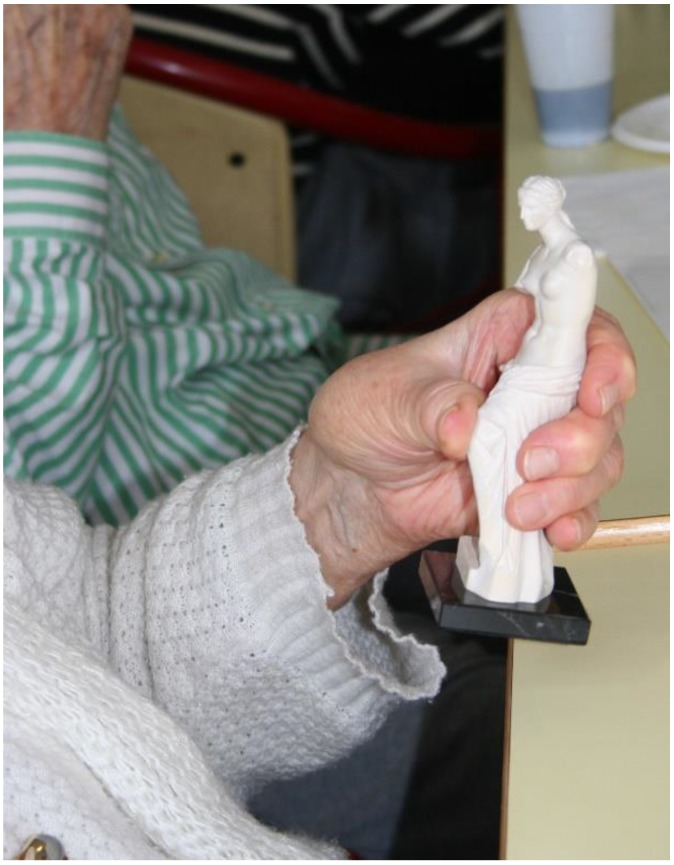
Art guided discussion about the Venus de Milo.

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
