# Peer review of "Museum Moving to Inpatients: Le Louvre à l’Hôpital"

_ijerph, 2019, doi:10.3390/ijerph16020206_

Reviewer 1 Report

This is the report of the unique activity which was performed with cooperation of the Louvre. As everyone knows, the Louvre is worldwide evaluated museum, that's why I think this report is valuable. 

But there are several points which must be improved with more explanations and discussions.

Minor comments:

1. You should give some information how did you prepare for this activity. It will be helpful for many people in other hospital who want to carry out a similar activity by the cooperation of other art museums. For example, what did you do for the copyright on making copies and the replicas of the art? Who is responsible for making copies? What kind of procedure was necessary in the Louvre and in the hospital? How much did it cost to perform this activity, and who paid for it? etc.

2. It is said that "Patients were asked qualitatively for their perception of decreased anxiety after completion of the session" and "Patients were asked for satisfaction." Please show the concrete items of the questionnaire.

3. This art therapy program was held in the department of rehabilitation, medicine of addictions, long-term care geriatrics and palliative care. Were there any differences in responses of patients between these department? Which department showed the most effective result? And please show the objective data(if you have) and discuss what kind of diseases which patients have are suitable for this art therapy programs.

4. Recommend putting the term ‘limitation’ in Discussion. It is important to specifically note the lack of any type of control group.

Author Response

This is the report of the unique activity which was performed with cooperation of the Louvre. As everyone knows, the Louvre is worldwide evaluated museum, that's why I think this report is valuable. 

But there are several points which must be improved with more explanations and discussions.

Minor comments:

1.     You should give some information how did you prepare for this activity. It will be helpful for many people in other hospital who want to carry out a similar activity by the cooperation of other art museums This has been included in the revised Msc. We did’nt include the cost of the replicas since they are part of the Museum’s collections and serve to other purpose after they have been removed from the hospital. The unique cost is : 1 high quality reprints of paintaing ; 2 the time spent by the Museum staff members and the paramedics. Both were employees of our 2 administrations (Page 2, lines 63-65). For example, what did you do for the copyright on making copies and the replicas of the art? Who is responsible for making copies? What kind of procedure was necessary in the Louvre and in the hospital? How much did it cost to perform this activity, and who paid for it? etc.This has been explained Page 3, linen 99-105

2. It is said that "Patients were asked qualitatively for their perception of decreased anxiety after completion of the session" and "Patients were asked for satisfaction." Please show the concrete items of the questionnaire. Questions have been added Page 3, line 111-113 and page 5, lines 165-169

3. This art therapy program was held in the department of rehabilitation, medicine of addictions, long-term care geriatrics and palliative care. Were there any differences in responses of patients between these department? Which department showed the most effective result? And please show the objective data(if you have) and discuss what kind of diseases which patients have are suitable for this art therapy programs. Page 5, lines 165-169 and page 5, lines 174-180.

4. Recommend putting the term ‘limitation’ in Discussion. It is important to specifically note the lack of any type of control group. Accordingly, a large limitation of the study section as been added Page 6, lines 227-240

Reviewer 2 Report

Thank you very much for reviewing the paper. 

Unfortunately, I am afraid to say this study is not appropriate for publication. 

Main reason is the study design is not scientific at all. 

Authors should explain more in detail why art intervention is effective to survivors of acute diseases and why hospital should engage art intervention to the patients as part of their program. 

Most of readers think some people enjoy art and others don't because their preferences are different. 

Authors should convince mechanism of links between art and trauma readers logically and psychologically. 

In methods, 

Statistical methods and possible confound variables should be taken into account. 

Characteristics of patients, more detailed explanation about program. 

Also, peer counseling might be the most effective part to heal the patients rather than art.. 

Hope these comments are helpful for revising the manuscript. 

Author Response

Comments and Suggestions for Authors

Thank you very much for reviewing the paper.                                    

Unfortunately, I am afraid to say this study is not appropriate for publication. 

Main reason is the study design is not scientific at all. I agree with the reviewer’s comment. We started our study late in the program. At the beginning, I had no idea at all of any scientific study about this experience. As anyone can see from pubmed medline, I’m a cardiologist and I previously published more structured manuscropts. However, during the Louvre program  we were all surprized by the results and started to collect as many data as we can. That’s the reason why many other are lacking. Also my feeling was that this museum experience is interesting enough to be proposed to a broad readership, even in an incomplete state of achievement

Authors should explain more in detail why art intervention is effective to survivors of acute diseases and why hospital should engage art intervention to the patients as part of their program. We don’t believe hospitals should engage art interventions, but we  though instead  that in some circumstances they may do it when a partnership with any art museum is easily suitable, or when a museum is interestd ec Page 6, lines 205-212

Most of readers think some people enjoy art and others don't because their preferences are different. This suggestion was added to the revision Page 6, lines 238-239

Authors should convince mechanism of links between art and trauma readers logically and psychologically. Data from previous studies were listed accordingly in the introduction section, in the beginning of the discussion section, as well as an additional comment by S Zeki et al. Page 6, lines 205-220 and 228-235.

In methods, 

Statistical methods and possible confound variables should be taken into account. Same comment as above

Characteristics of patients , more detailed explanation about program.Pages 2 and 3, lines 62-105 

Also, peer counseling might be the most effective part to heal the patients rather than art.. This has been added in the revised submission Page 6, lines 228-235

Reviewer 3 Report

Depressive symptoms are common in many critically ill patients. And although they has been recognized as the main form of emotional distress, their prevention and treatment in those patients is often overlooked. Art therapy is a well-known modality of spiritual support therapy and complementary medicine. The study submitted to review reports a new approach in which museum moves to the hospital and to the patient himself by displaying and discussing on an interactive basis artworks in patients’ rooms, dining rooms, circulation areas, and hospital ‘s garden.

Overall evaluation:

·         the title represent manuscript's contents, however, if the authors wanted to introduce a component to the title in French, they should have written Museum moving to inpatients: Le Louvre a l’hopital.

·         it is within the scope of the journal

·         the Abstract is compendious

·        presented work contain 7 typewritten pages and 23 references, properly chosen and thematically related to the content of the paper

·         methods are properly described

After reading the manuscript, I cannot disagree with the Authors who wanted treat patients with depression, with the art intervensions. However, art as a modality is nothing new. New is the approach itself- moving museum to the hospital. And it is also not surprising for me that patients wanted to speak about their perceptions of the artwork, the reasons for their choices and the personal life-recollections they advocated about. At least, someone finally devoted them his own time and let them express deeply hidden emotions (which art helped).

The obvious limitation of this study is that we cannot predict the long-term effect of this form of therapy because the Authors did not conduct a follow-up assessment after the art therapy had ended. And we do not know if trained paramedics will have time and willingness to continue art interactive discussions.

It would be also interesting to arrange similar studies, with discussions, but without artworks (a kind of control group), to show that it is art, not the personal relations and conversations themselves, that have a healing effect on the patients.

And minor concerns:

There are some mistakes in the work, e.g.:

®    page1, line 44: has be shown – should be has been shown;

·         there is discrepancy between the number of trained healthcare givers- page 3, line 111 – 160 and page 4, line 144 – 162;

Author Response

Comments and Suggestions for Authors

Depressive symptoms are common in many critically ill patients. And although they has been recognized as the main form of emotional distress, their prevention and treatment in those patients is often overlooked. Art therapy is a well-known modality of spiritual support therapy and complementary medicine. The study submitted to review reports a new approach in which museum moves to the hospital and to the patient himself by displaying and discussing on an interactive basis artworks in patients’ rooms, dining rooms, circulation areas, and hospital ‘s garden.

Overall evaluation:

·         the title represent manuscript's contents, however, if the authors wanted to introduce a component to the title in French, they should have written Museum moving to inpatients: Le Louvre a l’hopital. This has been madePage 1, line 2

·         it is within the scope of the journal

·         the Abstract is compendious

·        presented work contain 7 typewritten pages and 23 references, properly chosen and thematically related to the content of the paper

·         methods are properly described

After reading the manuscript, I cannot disagree with the Authors who wanted treat patients with depression, with the art intervensions. However, art as a modality is nothing new. New is the approach itself- moving museum to the hospital. And it is also not surprising for me that patients wanted to speak about their perceptions of the artwork, the reasons for their choices and the personal life-recollections they advocated about. At least, someone finally devoted them his own time and let them express deeply hidden emotions (which art helped). Thanks for this comment that has been taken into account. We added the response in the limitation of the study section (art-induced personal recollections are associated with less psychological potential trauma ?)Page 6, lines 228-235

The obvious limitation of this study is that we cannot predict the long-term effect of this form of therapy because the Authors did not conduct a follow-up assessment after the art therapy had ended. And we do not know if trained paramedics will have time and willingness to continue art interactive discussions.Page 6, lines  235-239

It would be also interesting to arrange similar studies, with discussions, but without artworks (a kind of control group), to show that it is art, not the personal relations and conversations themselves, that have a healing effect on the patients. Discussed also in the Limitation Section Page 1, lines 39-50, Page 6, lines 228-235

And minor concerns:

There are some mistakes in the work, e.g.:

®    page1, line 44: has be shown – should be has been shown; Done

·         there is discrepancy between the number of trained healthcare givers- page 3, line 111 – 160 and page 4, line 144 – 162;Correction made. Thank you

Round  2

Reviewer 1 Report

The contents have been improved into which readers can be impressed  by the significance of an art museum coming to the hospital . I evaluate the faithfulness of the author to have corrected the article responding the requests of the reviewer.

Reviewer 2 Report

Dear authors,

I have read the comments and revised versions. Thank you very much.

I am afraid to conclude the manuscript is not publishable in IJERPH. 

I also agree this program would be helpful to patients. However, the effect of the intervention should be proved scientifically. 

In introduction, there may be more previous interventions psychologically for traumatized survivors who experienced acute disease. Reading these previous articles may be helpful to the authors to explain what contents (example. peer counsering, going out, or ) of their interventions are effective psychologically. 

In methods, 

For measurements, reading previous articles may be helpful for authors to write about the measure scientifically. The number of items, anchor of answer, range of total score, factors, their validation and reliability of these measures should be stated in measurement sections. 

For statistical methods, 

Although authors did not use any statistical methods (only using number and percentage), authors should start from reading basic knowledge of statistic books. 

Thank you.

Reviewer 3 Report

I have read the above article - corrected version, the suggested amendments were included, so the manuscript is now suitable for publication in International Journal of Environmental Research and Public Health.